# Investigating Concrete Properties Using Dielectric Constant from Ground Penetrating Radar Scans

**Jonathan M. Taylor and Isabel M. Morris \***

Department of Civil and Environmental Engineering, New Mexico Institute of Mining and Technology, Socorro, NM 87801, USA
* Correspondence: isabel.morris@nmt.edu

**Abstract:** Determining the material properties and existing capacity of concrete infrastructure using nondestructive techniques is challenging due to evolving design requirements, unknown as-built conditions, and ongoing maintenance and deterioration. Concrete's material properties, including density, porosity, and compressive strength, are usually determined by mechanical testing, but being able to measure these properties noninvasively could aid engineers in maintaining and designing concrete infrastructure. Research into nondestructive methods for determining material properties of concrete has shown relationships between mechanical properties and ground penetrating radar (GPR) properties such as dielectric constant, attenuation, and instantaneous amplitude. We investigated direct relationships between dielectric constant and the density, porosity, and compressive strength of 23 mature concrete samples with varying mix designs using a commercial 1 GHz GPR. In normal-weight concrete, weak trends were seen between a dielectric for compressive strength ($R^2 = 0.76$) and one for density ($R^2 = 0.64$), whereas no significant trend was found with porosity ($R^2 = 0.52$). The GPR unit used provides acceptable data but has limited resolution for analyses and utility. The dielectrics showed distinct clustering by mix type—particularly the inclusion of materials such as blast furnace slag. While demonstrating that the dielectric constant is a candidate for rapid concrete comparisons, there is also a demonstrated need for further investigation of the complex relationships between mechanical and electromagnetic properties in concrete.

**Keywords:** ground penetrating radar; dielectric constant; concrete; compressive strength; density; porosity; nondestructive material testing

## 1. Introduction

The current state of the United States' infrastructure is in disrepair [1]. As the need for new infrastructure grows, the need to classify and determine the stability of existing infrastructure grows. Nondestructive testing is a group of methods to classify and inspect the current state of infrastructure components without having to remove materials to conduct laboratory tests [2]. In current construction and surveying, ground penetrating radar (GPR) is a commonly used tool to inspect structures for obstructions to work or to locate buried features [3]. GPR is the application of microwave radar pulses to measure the responses of materials. A pulse is emitted from an antenna into a material where the wave will pass through and reflect back at another antenna at varying speeds and intensities. The velocity of the wave is proportional to the dielectric constant in most construction materials. The shape, location, and strength of the reflected signal can be used to model changes in the physical structure or composition, along with other properties of a material, such as density and moisture content in the case of soil [4]. The sampling frequency and GPR frequency can be selected depending upon the application of the tool. Higher frequency GPR waves provide more resolution for small differences material properties but have a shallower maximum depth, whereas lower-frequency GPR antennas have less resolution within a material but will propagate to a greater depth. Sampling frequency will

be higher in analytical GPR units, with which data are expected to be pulled from the traces themselves rather than use visual inspection, as in some commercial GPR units tailored to certain applications. In concrete applications, frequencies between 0.9 and 2 GHz are used regularly.

For concrete applications, GPR is most commonly used to detect corrosion of rebar and cracking, and in determining the locations of objects within the concrete. Some current work is focused on the relationships between GPR properties and concrete's properties (e.g., [5–8]). GPR attributes such as instantaneous amplitude, intensity, and phase have been used to begin to describe relationships and make predictions of density, porosity, and compressive strength [5]. The GPR attributes could be used to predict material properties with some success—porosity and density being predicted better than compressive strength [5]. In [6], GPR attenuation and dielectric constant were used to model moisture content and chloride content in concrete. GPR has also been used to determine dielectric constants and relate them to concrete hydration, but material properties were not directly related to the GPR attributes [8]. In [7], dielectric constant and wave energy and their relationships with mix designs were investigated, showing that the water–cement ratio most strongly affected the permittivity due to the high dielectric of the non-reacted water compared to the relatively low dielectric of hydrated concrete [7]. Reference [7] also showed a trend of decreasing permittivity to strength for newly cast concrete samples following the hydration relationships previously discussed. In other work, a polynomial model for determining a dielectric constant with a basic mix design was produced for saturated concrete and dry concrete that had hydrated for at least 28 days [9]. The model showed that the dielectric constant in dry concrete was influenced heavily by the nature of the aggregate, the cement type, and the initial water–cement ratio. Chung et al. [10] correlated the dielectric constant with the compressive strength of 28-day concrete samples using a range of GPR frequencies from 10 MHz to 6 GHz. The research showed the dielectric constant increasing with compressive strength at all frequencies [10]. The two-way travel-time method is most commonly used for determining the dielectric constant; it tends to be favored over geometric hyperbola fitting because it is less prone to human error and requires less manual computation [6,7]. For concrete inspection, high frequency antennas dealing with frequencies between 1 and 2 GHz, and sometimes at 900 MHz, are preferred for their resolution and adequate depth penetration for most concrete investigations [5–8]. In these previous studies, the dielectric has rarely been studied for direct relationships with material properties; [7,8] used the dielectric to track hydration and moisture in young concrete. Previous work by Morris et al. [5] did not include the dielectric as a considered attribute, and Senin et al. [6] used dielectric and radar amplitude attenuation to model moisture and chloride content in concrete slabs. Previous work has also concentrated on GPR modeling of new cast concrete rather than providing insight into mature, in situ concrete [7].

This paper seeks to further investigate the relationship between the dielectric constant and material properties of fully hydrated concrete in order to help further the field of nondestructive testing. Compressive strength, density, and porosity are common properties used to classify concrete, and trends have been seen between dielectric constant and strength [7,11]. The work correlates dielectric constants calculated from GPR scans directly with the previously mentioned mechanical properties. The dielectric constant is targeted based on previous research documenting other GPR attributes having measurable relationships with material properties (e.g., [5–7,10]). Specifically, this paper builds on the previous work in [5] predicting the material properties of concrete with dielectric constants only.

## 2. Materials and Methods

### 2.1. Concrete Samples and Tests

Fifteen concrete beams cast using 23 different concrete mixes were reused in this work; the mixes and beams are described in more detail in [5]. Samples were 150 mm in width,

175 mm deep, and 1 m long and had number 4 sized rebar running centrally through each beam (Figure 1).

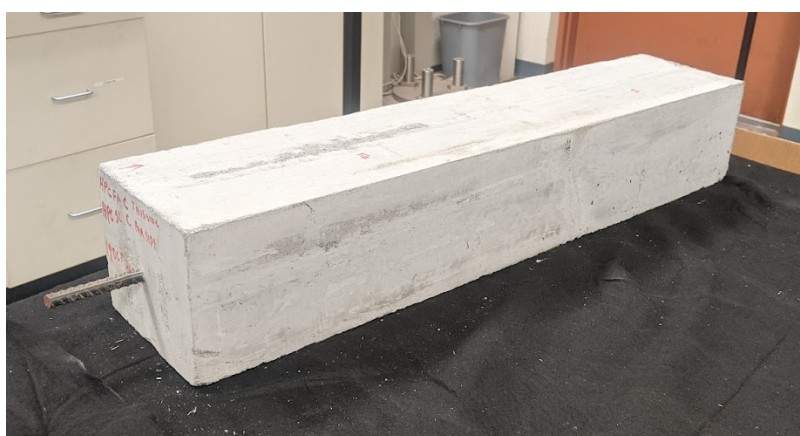

**Figure 1.** Example of a concrete beam sample; note central rebar.

The concrete mixes used various aggregates, additives, curing conditions, and cements to achieve varying material properties (Table 1). The base mix refers to the mix design used, being normal-weight concrete, high-performance concrete (HPC), and alkali-activated concrete (AA). The additives refer to supplementary cementitious materials (SCM) added to the base mix designs. The three curing conditions were used to vary the moisture available during the process, including being exposed to open air, covered, or cured at constant relative humidity.

**Table 1.** Concrete beam sample details; curing conditions were exposed (E), covered (C), and kept at constant relative humidity (H).

| Base Mix | Normal Weight Concrete | | | | | | | | | | | | |
|---|---|---|---|---|---|---|---|---|---|---|---|---|---|
| Additives | Flyash (FA) | | | Slag (SL) | | | None (CF) | | | Paste Only (P) | | | |
| Curing Condition | E | C | H | E | C | H | E | C | H | E | C | | H |
| w/c ratio | 0.41 | | | 0.41 | | | 0.41 | | | 0.41 | | | |
| average density (pcf) | 150 | | | 149 | | | 150 | | | 132 | | | |
| average porosity | 0.048 | | | 0.049 | | | 0.062 | | | 0.098 | | | |
| average strength (MPa) | 46.6 | | | 35.8 | | | 43.0 | | | 55.8 | | | |
| **Base Mix** | **High Performance Concrete** | | | | | | | | | **Alkali Activated Concrete** | | | |
| Additives | Flyash (FA) | | | Slag (SL) | | | None (CF) | | | N/A | | | |
| Curing Condition | E | C | H | E | C | H | E | C | H | E | C | | H |
| w/c ratio | 0.34 | | | 0.34 | | | 0.28 | | | 0.40 | | | |
| average density (pcf) | 152 | | | 150 | | | 153 | | | 136 | | | |
| average porosity | 0.039 | | | 0.036 | | | 0.022 | | | 0.099 | | | |
| average strength (MPa) | 37.6 | | | 37.3 | | | 53.8 | | | 34.2 | | | |

Material properties, including compressive strength, density, and porosity, were measured for the full set of concrete samples using conventional testing methods for concrete cylinders. Compressive strength was determined using ASTM C 39 and porosity using the cold-water saturation method [12,13]. In this work, GPR scanning was performed on the mature samples for the purpose of determining the dielectric constant of each sample.

### 2.2. GPR Scanning

A Conquest 100 GPR machine from Sensors and Software (Figure 2) was used in all GPR scanning. The Conquest 100 was chosen due to its common usage in the field for concrete investigation; the commercial purpose of this instrument leads to limited user control over settings such as step size and sampling rate. The machine is a 1 GHz monostatic GPR antenna with a 10 GHz sampling rate, providing approximately 0.1 ns resolution and recording for a time of 5 ns. The device has a fixed trace spacing of 10 mm between traces. All beams were placed adjacent to each other, allowing a single-line scan perpendicular to the rebar (Figure 2). Perpendicular scans were used to obtain hyperbolas for dielectric-constant calculation. From these scans, the dielectric constant was determined by both geometric hyperbola fitting and two-way travel-time calculation. To assess the correlation between dielectric constant and material properties, repeated scans were taken of the beams at constant temperature and humidity. A further set of scans on a single beam was used to determine the repeatability of the two-way travel-time method for velocity calculation.

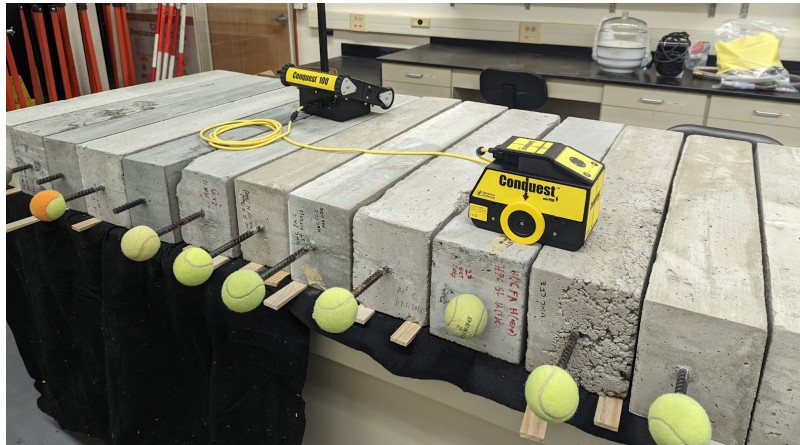

**Figure 2.** Single-line scanning setup showing adjacent beam placement.

### 2.3. Data Processing

The data were initially handled by converting the Sensors and Software GPR file formats (.dt1 and .gpz) into a CSV format using EKKO v6. CSV files could then be easily read into MATLAB as matrices. A MATLAB script was then used to split the scan data into individual matrices, one for each beam.

Postprocessing was used sparingly in this project due to data resolution and sample sizes. A dewow was applied to the traces with no gain applied. The two-way travel-time method for determining velocity is amplitude independent, allowing postprocessing to adjust amplitudes of peaks as long as their positions in time were preserved. To ensure accurate data analysis, the peaks located using the algorithm were confirmed visually in the traces to be the first peak of the initial pulse from the GPR and the first peak from the rebar reflection. Figures from EKKO are wiggle plots with positive/right side peaks filled.

Velocities were determined two ways, using the two-way travel time and hyperbola fitting using EKKO v6 software. For two-way travel times, the traces were plotted as contours in MATLAB (Figure 3). The plot of each beam's traces was used to confirm that the correct peaks, and timings were used for velocity calculations. The script calculated a velocity from every trace by locating the first pulse peak from the GPR and the first reflected peak from the rebar. The minimum time traveled would be found from the trace at the closest location to the rebar, and this time was used to calculate velocity and dielectric. The ground truth distance used in the velocity calculation was taken to be the measured

distance to the rebar from the top surface of the concrete beam. The velocity for two-way travel time was calculated using Equation (1).

$$v = 2 * \frac{d}{t_{peak} - t_{initial}},$$ (1)

where $v$ is the velocity, $d$ is the measured distance from the beam surface to the rebar, $t_{peak}$ is the time recorded at the peak of the hyperbola, and $t_{initial}$ is the time recorded at the first peak of the initial GPR pulse.

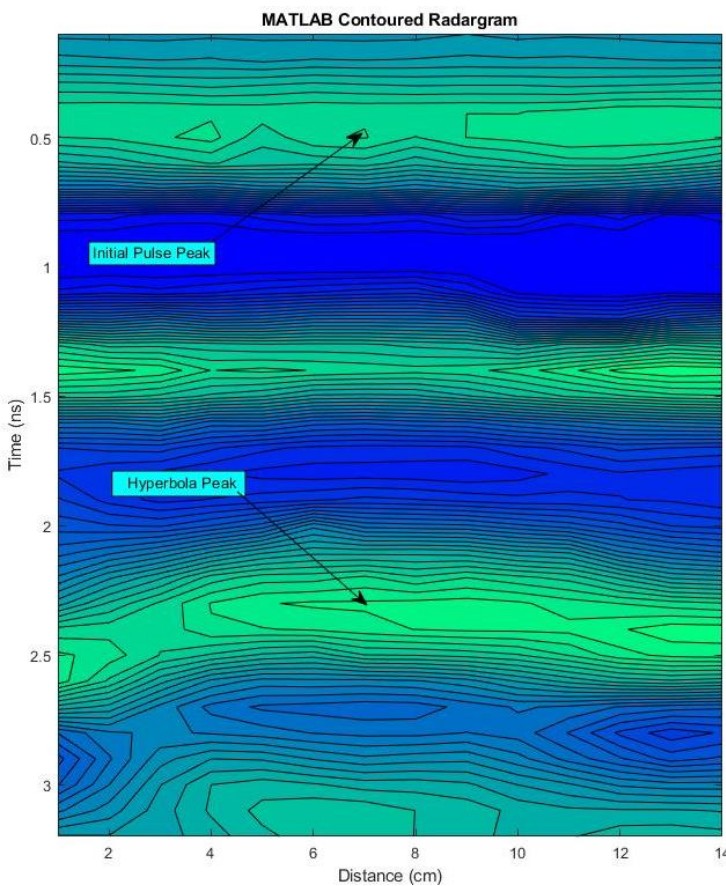

**Figure 3.** Radargram of single beam section shown in MATLAB using a contour plot. Note peak of hyperbola around 7 cm and 2.25 ns depth.

Velocities were determined from hyperbola fitting using Equation (2) [4].

$$t^2 - \frac{4}{v^2}x^2 = t_o^2,$$ (2)

where $t$ is the time of travel to a point on the hyperbola, $x$ is the location of the antenna from the origin of the hyperbola, $v$ is the velocity of the GPR scan in the material, and $t_0$ is the time location of the hyperbola peak. The hyperbola fitting procedure is performed manually in EKKO v6 on the radargrams, fitting a hyperbola to the rebar in each beam in the scan (Figure 4). Fitting was subject to user error, as the peak of the true hyperbola may be between or on top of the peaks in the traces available.

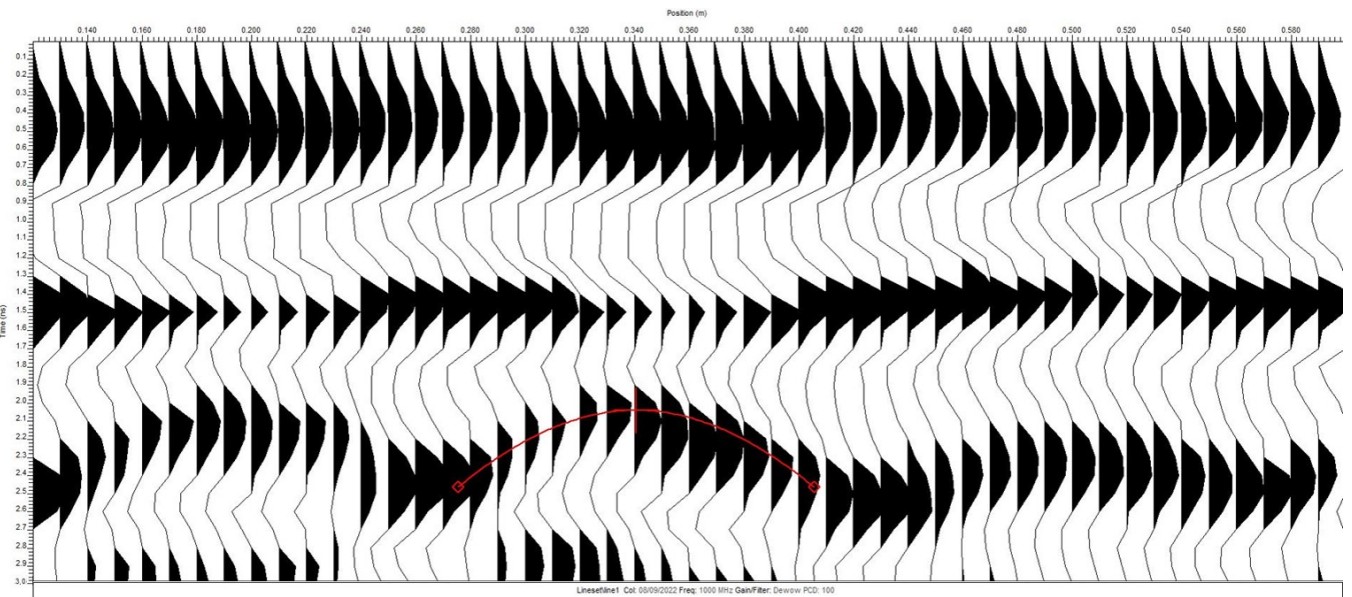

**Figure 4.** Radargram of continuous scan in EKKO V6, with one hyperbola shown. Note the different hyperbolas from the rebar in each beam.

For both methods, the calculated velocity is related to the dielectric constant by Equation (3) [4]:

$$\varepsilon = \left(\frac{c}{v}\right)^2, \tag{3}$$

where $c$ is the speed of light, $v$ is the velocity of the GPR scan in the material calculated by two-way travel time, and $\varepsilon$ is the dielectric constant of the material.

Once dielectrics were found for each mix, regressions were performed using matrix linear regression fitting a linear polynomial model between the dielectric and mechanical properties. $R^2$ values were calculated for each regression model. The sample's corresponding density, porosity, and compressive strength were used independently for the purpose of making in situ predictions of each mechanical property from their relationships with GPR properties. For correlation against mechanical properties, the data were plotted against each mechanical property and distinguished by one of the following—base mix, additives, or curing condition—to inspect for any trends caused by these factors [5].

*2.4. Repeatability*

The repeatability and variance of the scanning and calculation methods were determined by collecting a set of 50 scans of a single beam. All scans were taken perpendicular to the rebar at varying points along the beam. These data were then processed using the same methods to determine the dielectric constant; only the two-way travel-time method was used to help determine the error caused by either the GPR unit without human interpretation. From the velocities, a mean and standard deviation were calculated for the dataset, along with the error caused by the texture of the rebar.

**3. Results**

The repeatability testing showed that for a single mix, the travel time calculated using the MATLAB script method had low variability (Figure 5). The dielectrics had low variance (0.26 or 1.5%) using the two-way travel method in the 50 scan repeatability testing. The data had a standard deviation of dielectrics of 0.51 and a mean dielectric constant of 7.87. The GPR machine has a resolution of 0.1 ns, which is reflected in the stability scanning as the variance in the travel times. In Figure 5, the difference between the modal score and the maximum and minimum scores is 0.1 ns, meaning that the largest source of identifiable error is the GPR sampling rate. The lower resolution in the traces may have caused peaks

to lie on either side (±0.1 ns) of the actual peak location. Another source of error was the texture on the rebar affecting the travel distance of the pulse. By using the depth to the top of the rebar texture and the bottom of the rebar texture in Equation (1), we could determine the effect of this texture on the dielectrics. The difference in dielectrics due to rebar texture was 0.2, which was much less than the standard deviation of the calculated dielectric constants (0.51), showing the texture was not a significant source of error in these tests. This indicates that the resolution of the GPR likely causes the most error in dielectric-constant calculation, as opposed to operator error, conditions of the concrete surface, or the texture of the rebar.

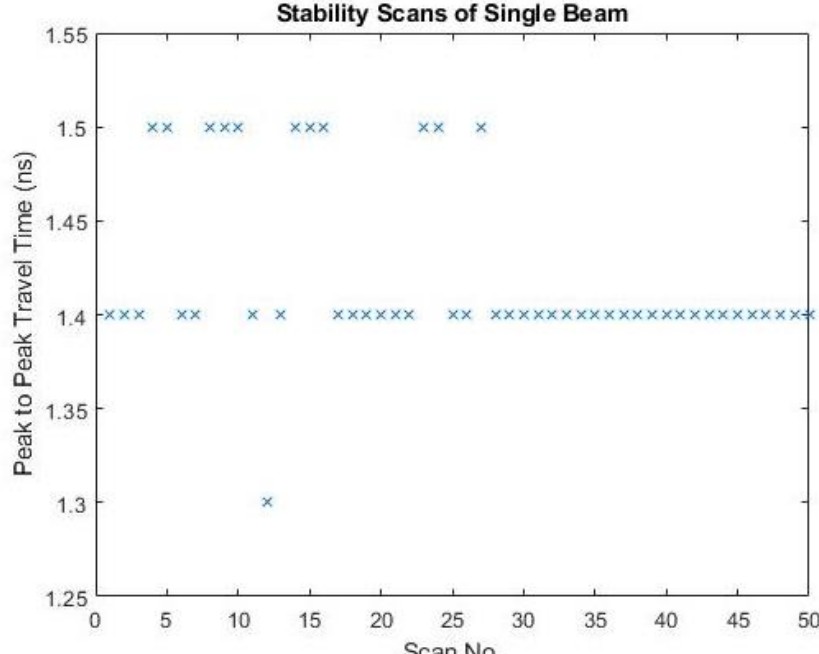

**Figure 5.** Plot of travel times to rebar in 50 scans of a single concrete beam. Note the resolution of 0.1 ns in the values.

The two-way travel time is subject to less human error than hyperbolas so the two-way travel time values were used for further correlation analysis. A correlation plot was found for the hyperbola and the two-way travel time to assess the difference between two-way travel time and hyperbola fitting for dielectric calculation (Figure 6). A larger range of dielectrics was found using the two-way travel-time method than with the hyperbola fitting. A positive correlation was found but was not strong enough to support the use of hyperbola fitting. This correlation between the two methods for determining dielectric constant may be improved by using an antenna with a higher sampling rate to give clearer peaks for hyperbola fitting or better two-way travel time calculation.

In the additive-distinguished plots (Figures 7b–9b), the alkali-activated concrete is not plotted, as it contained flyash, silica fume, and slag with no ordinary Portland cement (Table 1). For all mechanical properties, the base mix had the strongest apparent trend, so regression was performed on this subset. In the case of the normal-weight concrete, regressions were performed with and without the paste samples, which contained only sand (fine aggregate) and no coarse aggregates. In the case of compressive strength, the aggregate is an important factor [9]. No regressions were conducted on the alkali-activated concrete samples because only three points are available. It can be noted that alkali-activated concrete was distinguishable from the high-performance concrete and the normal-weight concrete in all cases by having the highest dielectric. The high dielectric found for this mix could have been due to the more conductive chemicals seen in the

concrete, which contained large amounts of SCM such as blast furnace slag (11.42% by weight) and class F/C flyash (11.42% by weight).

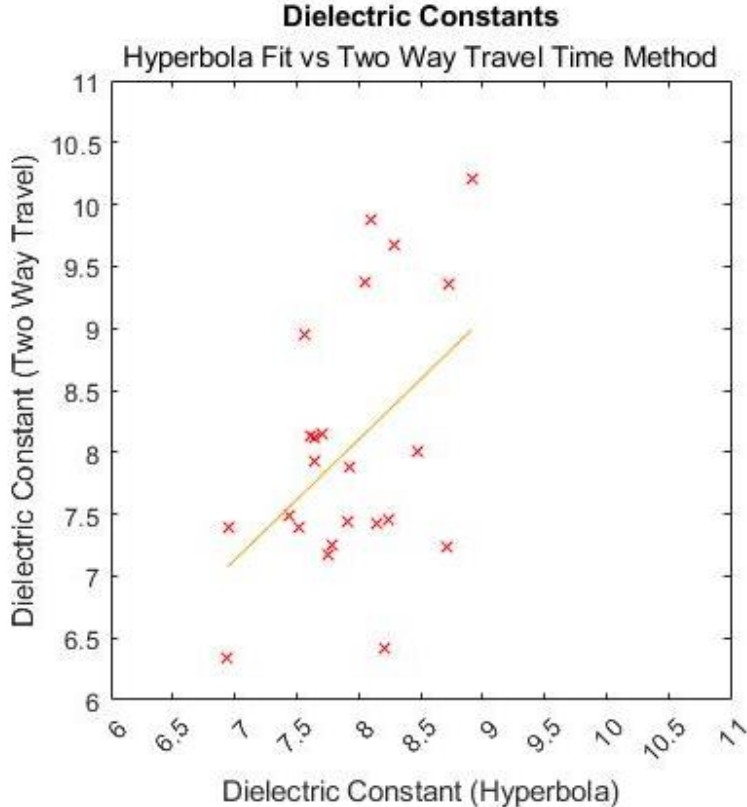

**Figure 6.** Correlation plot of dielectric constants found with hyperbola fitting and two-way travel time.

When the dielectric is plotted against compressive strength, it can be seen that the dielectric constant increased with compressive strength for the normal-weight concrete (Figure 7). No trend is noticeable for the high-performance concrete or the alkali-activated concrete. For compressive strength, the linear regression with dielectric constant has an $R^2$ value of 0.05 for the high-performance concrete, 0.76 for the normal-weight concrete without the paste, and a value of 0.00 for the normal-weight concrete with the paste. Paste values were excluded for compressive strength because the coarse aggregate plays a large role in material properties, so this mix has a significantly different composition.

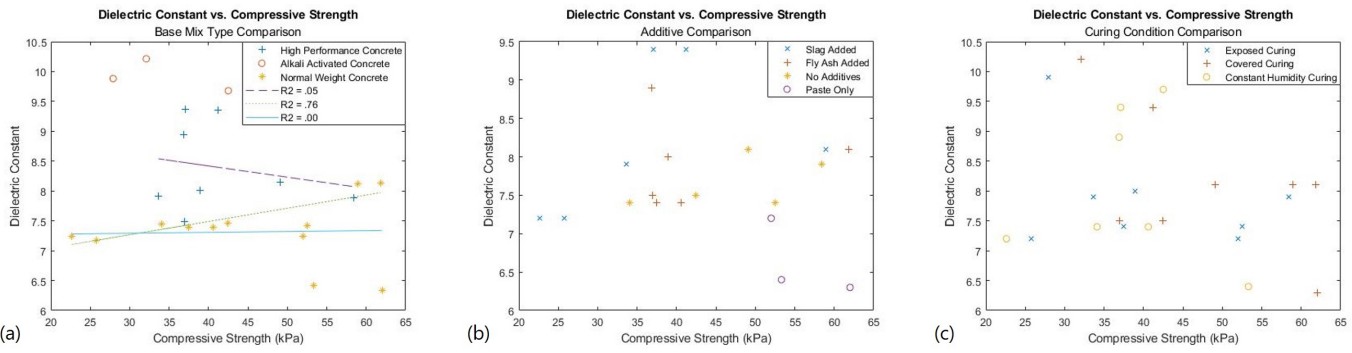

**Figure 7.** Dielectric constant vs. compressive strength (kPa) with breakdowns of base mix (**a**), additives (**b**), and curing condition (**c**).

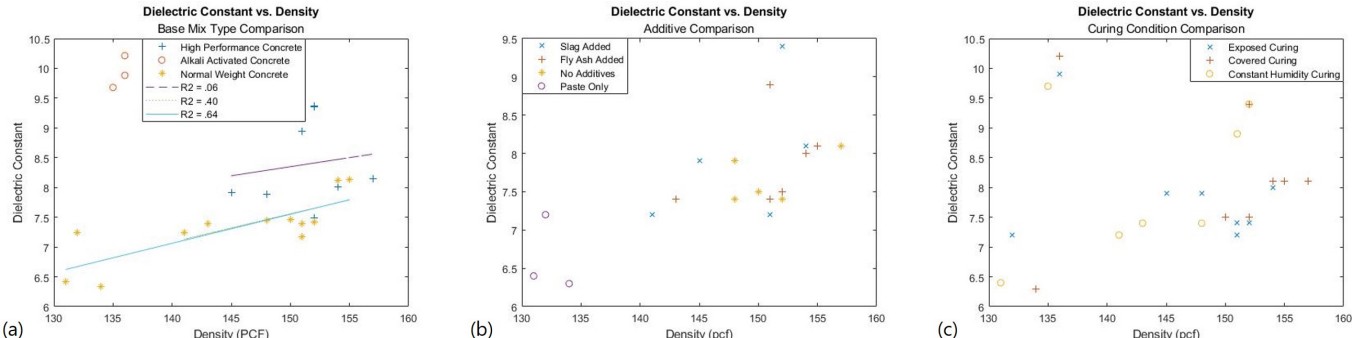

**Figure 8.** Dielectric constant vs. density (pounds per cubic foot, pcf) with breakdowns of base mix (**a**), additives (**b**), and curing condition (**c**).

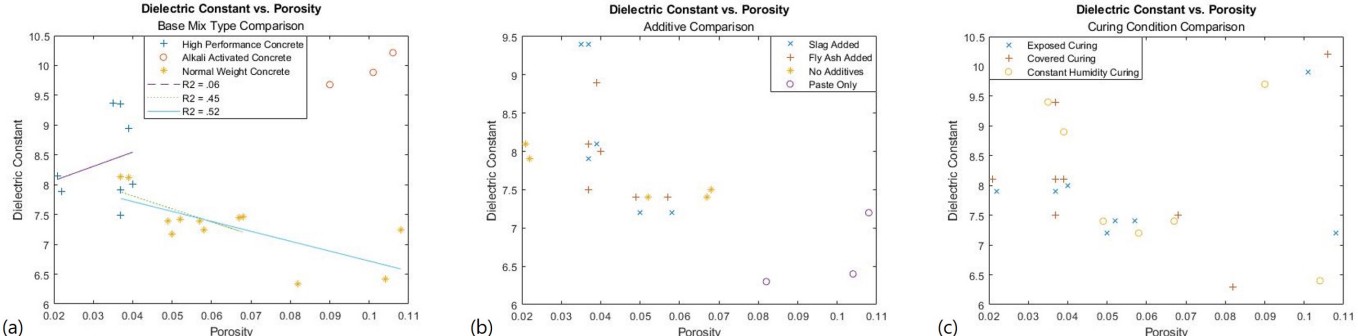

**Figure 9.** Dielectric constant vs. porosity with breakdowns of base mix (**a**), additives (**b**), and curing condition (**c**).

When the dielectric constants are plotted against density, we can see that dielectric constant increases with density for each base mix (Figure 8). The regressions for normal-weight concrete with and without the paste are almost exactly the same. The $R^2$ of the high-performance concrete was found to be 0.06. For the normal-weight concrete, $R^2$ of 0.64 and 0.40 were found with and without the paste, respectively, indicating that the coarse aggregate has little effect on density and should be included with density correlations; unlike compressive strength, the inclusion of coarse aggregate has little effect on the density.

Figure 9 shows that dielectric constant tends to decrease with an increase in porosity for the normal-weight concrete; no correlation can be seen for the high-performance concrete. The $R^2$ for the normal-weight concrete were 0.52 and 0.45 with and without the paste, respectively. This trend was expected due to the higher air content in the concrete at higher porosity, which should reduce the dielectric constant. Air (in the pores) has a dielectric of one, whereas concrete has a nominal dielectric of about seven; as more of a single material appears in a composite, the dielectric of the whole sample will shift towards the value of the dielectric of that material [4,14].

## 4. Discussion

This study further classified the relationships between the dielectric constant and material properties. For all properties, the base mixes are distinguishable from each other. This could lead to dielectric constants being used for mix classification, which was expected from the research into concrete dielectric calculation by [9]. As we continue this investigation with larger datasets, we expect the correlations and relationships between the dielectric constant and material properties to grow more apparent for each base mix due to the distinct groupings already seen.

The dielectric constant is an indicator of compressive strength, density, and porosity in normal-weight concrete. The trend seen for the normal-weight concrete matches the expected trend from [10]. We also found that the dielectric constant is sensitive to changes in

the SCM used in the concrete mix. For example, the heavy metals present in the slag and fly ash (present in the HPC and AA mixes) may have a stronger effect on the electromagnetic properties than on the physical properties. High-performance concrete and alkali-activated concrete will need more thorough testing to investigate the relationships that may exist between dielectrics and material properties. With the current mix designs, we do not know how the different mixes of SCM may be affecting the expected relationships. Future work investigating concrete with GPR will have to include investigating the effects of SCM and composition on dielectric constant, as in [9]. As such, future samples will be performed with systematic variation in mix design (including SCMs) to gain a clearer understanding of each material's effect on the dielectric constant and other properties.

The removal of the paste values increased the $R^2$ value in the case of the compressive strength. The coarse aggregate greatly affects the compressive strength of a sample, so it is recommended to remove paste dielectric constants from future datasets used to estimate compressive strength. In the cases of density and porosity, the coarse aggregate has little effect, so including the dielectric constant of it in future datasets is recommended.

The two-way travel-time method has proven to be more repeatable and useful for determining the velocity of GPR in concrete samples where ground truth depths are available. The hyperbola fitting method is far more limited by the step size resolution of the GPR antenna. Both methods would be improved by recording more closely spaced traces at higher sampling rates (in time), providing more peaks for the fitting and ensuring traces are taken while on top of the rebar and not while on either side.

## 5. Conclusions

To summarize the conclusions of this work:

- A relationship does exist between dielectric constant and material properties for normal-weight concrete.
- Dielectric constant is a weak indicator of compressive strength in normal-weight concrete ($R^2 = 0.76$).
- Density and porosity's relationships with dielectric constant are mostly inconclusive ($R^2 = 0.64$ & $R^2 = 0.52$).
- Not all mix designs are applicable to each material property relationship.
- Dielectric constant can be used to determine mix design type.
- The method is repeatable in a controlled environment.

The methods used in this paper and the possible applications of similar technologies being developed will be important for infrastructure inspection and the non-destructive testing fields. Adapting the method and technology for concrete inspection should be feasible, as GPR is such a well used technology already. The research presented in this paper will help to inform future research on the dielectric constant's relationship with the material properties of concrete mixes, including providing rational categories of concrete mixes that can be considered together for relationships with material properties. The possibility of rapid scanning and determination of material properties in concrete structures would aid engineers and contractors in ensuring adequate construction and resilient infrastructure. Further research will help develop new tools and methods for engineers and researchers to utilize to protect old structures and further the nondestructive testing field for civil engineering.

**Author Contributions:** Conceptualization, I.M.M. and J.M.T.; methodology, J.M.T.; software, I.M.M. and J.M.T.; validation, J.M.T.; formal analysis, J.M.T.; investigation, J.M.T. and I.M.M.; resources, I.M.M.; data curation, I.M.M. and J.M.T.; writing—original draft preparation, J.M.T.; writing—review and editing, I.M.M.; visualization, J.M.T.; supervision, I.M.M.; project administration, I.M.M.; funding acquisition, I.M.M. All authors have read and agreed to the published version of the manuscript.

**Funding:** This research was funded internally by New Mexico Tech.

**Data Availability Statement:** GPR data collected for this work are available on request from the corresponding author. The concrete mixes and physical property data are available in [5].

**Acknowledgments:** The authors would like to acknowledge Christopher Blue, Sam Upham, Link Patrick, Vivek Kumar, Isaac Morris, and Colin Morris for their help moving concrete samples.

**Conflicts of Interest:** The authors declare no conflict of interest. The funders had no role in the design of the study; in the collection, analyses, or interpretation of the data; in the writing of the manuscript; or in the decision to publish the results.

## Abbreviations

The following abbreviations are used in this manuscript:

| | |
|---|---|
| GPR | Ground penetrating radar |
| SCM | Supplementary cementitious materials |
| AA | Alkali-activated concrete mix |
| HPC | High-performance concrete mix |

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
