# Peer review of "Investigating Concrete Properties Using Dielectric Constant from Ground Penetrating Radar Scans"

_infrastructures, doi:10.3390/infrastructures7120173_

Round 1

Reviewer 1 Report

This paper is well- written and English language and style are fine. 

The methods being used seem to be accurate and lead to valid results. 

The conclusions are supported by the results. 

Finally , the content is satisfying.  

Author Response

The authors thank Reviewer 1 for helping with the manuscript.  From the comments received from the editor and reviewers, the discussion/conclusion section has been separated into two sections and the conclusions have been expanded. We also more explicitly stated in the conclusion that being able to compare concrete quickly and directly is useful for the field of civil engineering/construction.

Reviewer 2 Report

This is a good paper.  Interesting how you downloaded the amplitudes to just a spreadsheet and then used your own software to produce the reflection profiles (radargrams as you call them).  What would these look like in S&S software?  I'll bet the hyperbolas (which are barely visible in your analysis) would have nice long axes and be easier to pick and study.  But I see why you did it.  In this case you are only using RDP, or dielectric constant as you call it, as a measure of velocity, and then using velocity to get to the composition of the material.  That is basic GPR stuff, but perhaps in your field this is an important direct comparison for people to know and then apply quickly in the field?  If that is what you are doing, perhaps it might be better to come  right out and say that?  If it is something else, then I missed it..

Author Response

The authors thank Reviewer 1 for helping with the manuscript. From the comments received from the editor and reviewers, the discussion/conclusion section has been separated into two sections and the conclusions have been expanded. We also more explicitly stated in the conclusion that being able to compare concrete quickly and directly is useful for the field of civil engineering/construction, and that dielectric constant or velocity is a useful way to do this comparison. In addition, the final sentence of the abstract was edited to this effect: “While demonstrating that dielectric constant is a candidate for rapid concrete comparisons, there is also a demonstrated need for further investigation of the complex relationships between mechanical and electromagnetic properties in concrete.” Also note that S&S software (EKKO project) was indeed used for the hyperbola fitting - this is discussed on p.5 of the paper. The included figures of the wiggle plots in the paper (fig. 4) were our preference for performing the picking.